# Dried blood spot self-sampling at home is a feasible technique for hepatitis C RNA detection

Tamara Prinsenberg[1,2]*, Sjoerd Rebers[3], Anders Boyd[1], Freke Zuure[1], Maria Prins[1,2], Marc van der Valk[2], Janke Schinkel[3]

1 Department of Infectious Diseases, Research and Prevention, Public Health Service of Amsterdam, Amsterdam, The Netherlands, 2 Division of Infectious Diseases, Department of Internal Medicine, Amsterdam Infection and Immunity Institute, Amsterdam UMC, University of Amsterdam, Amsterdam, The Netherlands, 3 Department of Medical Microbiology, Section of Clinical Virology, Amsterdam UMC, University of Amsterdam, Amsterdam, The Netherlands

* tprinsenberg@ggd.amsterdam.nl

**Data Availability Statement:** All relevant data are within the manuscript and its Supporting Information files.

## Abstract

To facilitate HCV diagnosis, we developed an HCV-RNA testing service, which involved home-sampled dried blood spots (DBS). The main objective of this study was to evaluate the feasibility of self-sampling at home. Furthermore, to optimise the processing of DBS samples for RNA detection, we evaluated two elution buffers: phosphate-buffered saline (PBS) and L6-buffer. 27 HCV-RNA and 12 HIV-1 RNA positive patients were included. Laboratory spotted DBS (LabDBS) were made by a technician from blood samples drawn at inclusion. Patients received a DBS home-sampling kit and were requested to return their self-sampled DBS (ssDBS) by mail. We compared the RNA load of PBS and L6-eluted labDBS, and of L6-eluted ssDBS, L6-eluted labDBS and plasma. LabDBS load measurements were repeated after 7–13 and 14–21 days to evaluate RNA stability. All 39 plasma samples provided quantifiable RNA loads. In 1/39 labDBS sample, RNA could not be detected (plasma HCV load: 2.98 $\log_{10}$ IU/ml). L6-eluted samples gave a 0.7 $\log_{10}$ and 0.6 $\log_{10}$ higher viral load for HCV and HIV-1 respectively, compared to PBS-eluted samples. Strong correlations were found between labDBS and ssDBS HCV RNA (r = 0.833; mean difference 0.3 $\log_{10}$ IU/mL) and HIV-1 RNA results (r = 0.857; mean difference 0.1 $\log_{10}$ copies/mL). Correlations between labDBS and plasma values were high for HCV (r = 0.958) and HIV-1 (r = 0.844). RNA loads in DBS remained stable over 21 days. Our study demonstrates that self-sampling dried blood spots at home is a feasible strategy for the detection of HCV and HIV-1 RNA. This could facilitate one-step diagnostics and treatment monitoring in communities with high HCV prevalence.

## Introduction

In the Netherlands, transmission of hepatitis C virus (HCV) occurs primarily among HIV-positive men who have sex with men (MSM) as HCV incidence dropped to nearly zero among

**Funding:** This study was executed by the MC Free consortium (www.mcfree.nl) for the NoMoreC project (www.nomorec.nl). MC Free is funded by Gilead Sciences Europe, Gilead Sciences Netherlands, AbbVie, Janssen-Cilag, Merck Sharpe & Dohme, Roche Diagnostics Netherlands. The funders had no role in study design, data collection and analysis, decision to publish, or preparation of the manuscript. Gilead Sciences provided support in the form of the salary for TP, but did not have any additional role in the study design, data collection and analysis, decision to publish, or preparation of the manuscript. The specific roles of this author are articulated in the 'author contributions' section.

**Competing interests:** I have read the journal's policy and the authors of this manuscript have the following competing interests:TP, FZ and MP report speaker fees and grants from: Gilead Sciences, MSD and AbbVie paid to their institute. AB reports grants from ANRS and SIDACTION, outside the submitted work. SR has no relevant conflicts of interest to report. MvdV's institute received grants and speaker fees from: Abbvie, Gilead, Johnson & Johnson, MSD, ViiV, outside the submitted work. JS reports nonfinancial support from ROCHE Diagnostics, during the conduct of the study and grants from: Gilead Sciences, MSD, Abbvie, outside the submitted work. This does not alter our adherence to PLOS ONE policies on sharing data and materials.

people who inject drugs [1–3]. Since 2000, there has been an unexpected and substantial increase in HCV incidence among MSM [4, 5]. Most infections in this group are sexually transmitted and occur in human immunodeficiency virus-1 (HIV) infected MSM. However, recent studies have shown that HIV-negative MSM eligible for or on pre-exposure prophylaxis (PrEP) to prevent HIV acquisition are also at risk of acquiring HCV [6, 7].

In response to this epidemic, in 2016 we formed the Amsterdam MSM Hepatitis C Free consortium (MC Free) with the goal to stop HCV transmission among MSM in Amsterdam. In close cooperation with the MSM community, (commercial) stakeholders and health care professionals the NoMoreC project was developed. The project aims to increase the uptake of testing, improve the engagement of MSM in preventive behaviours and increase awareness among the population at risk and health care professionals. NoMoreC uses online and face-to-face interventions, including a training package, a prevention toolbox to assist risk reduction and a website (www.NoMoreC.nl). The project website offers information, videos and person-alized advice on risk reduction and testing options, including a HCV-RNA home-based testing service. This service involves self-sampling of dried blood spot (DBS) samples by a finger-prick, mailing the samples to our laboratory for testing and receiving the test result online. The NoMoreC home-based testing service allows MSM to test shortly after engaging in at risk activity, thereby enabling them to assume responsibility of their own health. Additionally, the service offers the possibility to test for HCV outside the health care system, which may lower the barrier to testing and increase testing frequency.

Previous studies have shown that DBS, spotted in the laboratory, can be reliably used for the detection and quantification of HCV-RNA, with reported detection limits varying between 2.2–3.4 $\log_{10}$ IU/mL [8–11]. Although the detection limits in DBS are 0.5–1.7 $\log_{10}$ IU/mL higher than in plasma, it is suitable for diagnosing patients with HCV infection, as most patients have viral loads >3.0 $\log_{10}$ IU/mL [9]. HIV-1 RNA testing on DBS has also been evaluated in several studies and was shown to have a limit of detection of 3.7 $\log_{10}$ copies/mL, 2 $\log_{10}$ higher than in plasma viral load assays [12].

As part of the NoMoreC testing service, DBS samples are transported by mail, which causes a delay in time between sampling and viral load measurements. As a consequence of unfavourable transport conditions, a decline of viral RNA on DBS over time could be an issue when stored at room temperature (RT). Reports on the stability of viral RNA on DBS are inconsistent: some studies observed a three to ten-fold decline in viral load after 3 days and 4 weeks of storage at RT [9, 13], whereas others showed that HCV-RNA levels on DBS remained stable at RT for over a year [14].

Only a limited number of studies have explicitly evaluated the use of self-sampled DBS (ssDBS) [15–20]. The majority of these studies reported the use of ssDBS for determining drug levels [16–19]. One large Japanese study showed the successful use of ssDBS sent by regular mail for HIV diagnosing HIV using an HIV antigen/antibody test [21].

To our knowledge, no studies have been performed to evaluate ssDBS for diagnosing a viral infection by measuring viral RNA. For a reliable HCV testing service, it is essential to assess if people can adequately sample a DBS at home and if HCV RNA can be detected in these samples without compromising sensitivity of RNA detection. Therefore, the main objective of the current study was to evaluate the self-sampling DBS technique for the detection of HCV-RNA. First, we assessed the quality of the ssDBS samples and the patients' experience with self-sampling. Second, we determined the agreement between laboratory spotted DBS (labDBS) and ssDBS viral load measurements and the correlation between labDBS and plasma viral loads. Third, elution buffers were compared to optimise the processing of DBS samples for RNA detection. Fourth, effect of RT storage on viral RNA levels was also assessed on LabDBS samples. Anticipating on future direction ssDBS may take for viral diagnostics, we also included

the detection of HIV-1 RNA in this evaluation. Finally, we studied the effectiveness of different elution buffers and explored the detection limit of HCVcAg in DBS.

## Patients and methods

### Patients

HCV or HIV-1 viremic patients were recruited during routine visits at the hepatology and infectious disease outpatient departments at the Amsterdam UMC, location Academic Medical Center, the Netherlands from September 2017 to April 2018. The study received ethical approval from the medical ethics committee of the Amsterdam UMC (Study nr. 2017_170), in accordance with the Helsinki declaration. After giving their informed consent participants received a DBS home sampling kit and 1 tube of EDTA whole blood was by a health care worker.

### Sample collection and processing

**ssDBS collection.** Finger-prick blood samples were collected by the participants at home with a DBS home sampling kit. The kit contained: paper instructions, 2 contact-activated lancets (2.0-mm BD Microtainer), 2 band aids, a DBS card with 5 circles (Whatman Protein Saver 903 card; GE Healthcare), an alcohol wipe, a gauze wipe, a grip seal bag, a desiccant sachet and a return envelope (UN 3373). To assist the blood collection, participants had access to an online instructional video (https://www.youtube.com/watch?v=AyDXoTkliLI) in addition to the paper instructions. Participants were instructed to fill all 5 circles on the DBS card, to air dry at RT overnight, and to write the sampling date on the card before putting it in a grip seal bag with the desiccant and return it by mail.

**User-friendliness of self-sampling.** Participants received a short questionnaire with the home sampling kit and were asked to rate their experience with DBS self-sampling. The questionnaire included yes/no questions and 5-point Likert scale questions about (1) the use and clarity of the instructions, (2) ease of performing a finger prick and (3) ease of making a blood-spot. The questionnaire and paper instructions were pre-tested by 3 patients and adjusted according to their feedback.

**ssDBS quality assessment.** On receipt in the laboratory, ssDBS samples were visually assessed by a laboratory technician using the quality criteria as described by Hoeller et al. [20]: spot size (circle completely filled), thoroughly soaked (observed from the back) and one application of blood (not composed of many small spots). In addition, the number of good spots were counted and the reasons for failed spots were recorded. All spots were subsequently processed regardless of their quality.

**Plasma and labDBS collection.** EDTA whole blood samples were collected by venipuncture by a health care worker and stored at -80°C. HCV and HIV-1 RNA testing was performed on plasma within 7 days. LabDBS samples were obtained by spotting 10 circles on 2 cards per patient (60 μL of EDTA whole blood per circle onto Whatman Protein Saver 903 card; GE Healthcare) and air dried overnight at RT. LabDBS cards were stored at RT in grip seal bags with desiccant until processing of the DBS sample at <7 days (t0), after 7–13 days (t1) and after 14–21 days (t2).

**DBS processing for testing.** Two spots were cut out of the DBS card with a clean pair of scissors. For each DBS sample, a new clean pair of scissors was used to avoid cross-contamination. Each spot was cut into small pieces (6 pieces/spot) to be able to fit in an Eppendorf tube and facilitate the elution process. The two cut up spots were placed into an Eppendorf tube (12 pieces/tube) and mixed with elution buffer. For the elution buffer comparison, only labDBS samples were used and eluted in both PBS (Phosphate Buffered Saline, BSA (10%) and Tween-

20 (0.05%)) and L6-buffer (500 g GuSCN in 91.7 mL 0.2M EDTA [pH8.0], 10.12 mL Triton X-100, and 416.7 mL 0.1M Tris-HCl [pH6.4]) at t0. Preparation of the buffers is described elsewhere [9, 22]. For the elution of ssDBS and the elution of labDBS at t1 and t2, only the L6-buffer was used. Two spots (120 μL of whole blood) were eluted in 800 μL buffer for HCV RNA measurement or in 1200 μL buffer for HIV-1 RNA measurement. The tubes were shaken (1000 rpm) for 1–2 hours at RT and then centrifuged for 5 minutes at 14000 rpm. The supernatant was transferred into a new Eppendorf tube and stored at -80˚C until viral load measurements.

## HCV RNA and HIV-1 RNA measurements

The CAP/CTM assay (COBAS Ampliprep/COBAS TaqMan; Roche Diagnostics) was used for extraction, amplification and quantification of HCV and HIV-1 RNA on plasma and DBS eluates from 2 spots. Briefly, 650 μL and 1000 μL plasma specimens and were pipetted into the specimen tubes, for HCV and HIV-1 measurements respectively. After vortexing the specimens were transferred to the CAP/CTM for processing. For DBS eluates, the same quantities (650 μL for HCV, 1000 μL for HIV-1) and procedures were followed.

## Statistical analysis

Simple descriptive statistics were used to analyse and report on the quality of the ssDBS samples and the user-friendliness of self-sampling.

HCV and HIV-1 RNA levels were $\log_{10}$ transformed. All analyses were run for HCV and HIV-1 separately. First, values obtained from labDBS and plasma were plotted and compared using Pearson's correlation. A linear regression model was then fit, using the plasma samples as the dependent variable and labDBS samples as the independent variable. For each pair of measurements, the differences between plasma and labDBS values were then plotted against their means in a Bland-Altman analysis. Assuming normal distribution of viral loads, the between-method difference and its limits of agreement (LOA) were calculated. The same was done comparing values obtained from ssDBS and labDBS.

Second, we examined the effectiveness of L6 and PBS buffer. LabDBS values measured in L6-eluates and PBS-eluates were plotted and compared using Pearson's correlation. A linear regression model was then fit, using the PBS buffer as the dependent variable and the L6 buffer as the independent variable. For each pair of measurements, the differences between values measured in PBS and L6-eluates were then plotted against their means in a Bland-Altman analysis. We then evaluated the diagnostic sensitivity of HCV and HIV-1 RNA detection in LabDBS and ssDBS, eluted in L6 buffer.

For those labDBS samples with repeated viral load measurements, the difference between plasma and labDBS values were calculated and plotted over time. The average change in difference over time was estimated by mixed-effect linear regression accounting for patient variability at initial sample using a random intercept.

Statistical analysis was conducted using STATA (v12.1, College Station, TX) and a *p*-value <0.05 was considered statistically significant.

## Results

A total of 39 viremic patients were included: 17 patients were HCV mono-infected, 11 were HIV-1 mono-infected and 11 were HCV/HIV co-infected. Of the co-infected patients, 10 had a detectable HCV viral load and undetectable HIV viral load (<40 copies/mL) and in 1 patient, both HCV and HIV-1 RNA were detectable.

**Table 1. Overview of study samples.**

| Target measured | Plasma samples n = 39 | | | | LabDBS samples n = 39 [α] | | | | ssDBS samples n = 34 [α] | | |
|---|---|---|---|---|---|---|---|---|---|---|---|
| | HCV mono-infected | HCV/HIV co-infected | Total | | HCV mono-infected | HCV/HIV co-infected | Total | | HCV mono-infected | HCV/HIV co-infected | Total |
| **HCV** | 17 | 10 | 27 | LabDBS made | 17 | 10 | 27 [α] | ssDBS received | 15 | 7 | 22 [α] |
| | | | | Eluted in PBS | 15 | 10 | 25 [α] | Eluted in PBS | 3 | 1 | 4 [α] |
| | | | | Eluted in L6 | 17 | 10 | 27 [α] | Eluted in L6 | 15 | 7 | 22 [α] |
| | HIV-1 mono-infected | HIV/HCV co-infected | Total | | HIV-1 mono-infected | HIV/HCV co-infected | Total | | HIV-1 mono-infected | HIV/HCV co-infected | Total |
| **HIV-1** | 11 | 1 | 12 | LabDBS made | 11 | 1 | 12 | ssDBS received | 11 | 1 | 12 |
| | | | | Eluted in PBS | 11 | 1 | 12 | Eluted in PBS | 3 | | 3 |
| | | | | Eluted in L6 | 10 | 1 | 11 | Eluted in L6 | 9 | 1 | 10 |

Total number of plasma samples drawn, DBS samples spotted in the laboratory (LabDBS) and DBS samples spotted by the patient (ssDBS). For the DBS samples the number of samples eluted by PBS and L6 buffer is shown. Numbers of samples of HCV and HIV-1 mono-infected and HCV/HIV co-infected patients are shown per target measured.

[α]: in 1 sample with a plasma viral load of 2.98 $\log_{10}$ IU/mL the target was not detected in DBS.

Table 1 gives an overview of the study samples; the total number of plasma samples, DBS samples spotted in the laboratory (labDBS) and DBS samples spotted by the patient (ssDBS). The number of DBS samples eluted in PBS and L6 buffer is also shown.

## ssDBS quality

In total, 34 of the 39 patients (87%) returned their self-sampled DBS, of which 33 (97%) were visually assessed according to the three quality criteria. Twelve out of 33 patients (36%) sampled 5/5 good quality spots that met all quality criteria and 19 patients (58%) sampled at least 2/5 spots meeting all criteria. The spots that did not meet all criteria were classified as failed. The main reason for a failed spot was that the circle was not completely filled (76% of failed spots).

## User-friendliness

Of the 34 participants whom returned their ssDBS sample, 33 had filled out a questionnaire regarding their experience with DBS self-sampling and the clarity of instructions (Table 2).

In total, 32 participants (97%), used the paper instructions and found these helpful. The instructions were rated as very clear by 15/32 participants, clear by 16/32 participants and very unclear by one participant. 17 participants (52%) used the instructional video in addition to the paper instructions and rated the video as clear (8/17) or very clear (9/17). The majority of participants (63%) experienced performing the finger as easy or very easy. Filling the circles correctly was experienced as easy (14/33) or very easy (2/33) by half of the participants and as hard (6/33) or very hard (2/33) by a quarter of the participants.

## Comparison of ssDBS and labDBS viral loads

Viral RNA was detected in 33 of 34 ssDBS samples. The negative ssDBS sample corresponded with the negative labDBS sample, with a plasma HCV RNA load of 2.98 $\log_{10}$ IU/mL. In total,

**Table 2. Patients' use and experience with DBS self-sampling kit.** Results of 33 returned questionnaires.

| | Nr. of patients (%) |
|---|---|
| **Use and clarity of instructions (N = 33)** | |
| **Number using paper instructions** | 32 (97%) |
| **Paper** instructions are: | |
| Helpful | 32 (100%) |
| Very clear | 15 (47%) |
| Clear | 16 (50%) |
| Very unclear | 1 (3%) |
| **Number using video instructions** | 17 (52%) |
| **Video** instructions are: | |
| Helpful | 15 (88%) |
| Not helpful | 2 (12%) |
| Very clear | 9 (53%) |
| Clear | 8 (47%) |
| **Experience with performing finger prick (N = 33)** | |
| Very easy | 11 (33%) |
| Easy | 9 (27%) |
| Not easy/not hard | 6 (18%) |
| Hard | 4 (12%) |
| Very hard | 1 (3%) |
| Not answered | 2 (6%) |
| **Experience with filling circles correctly (N = 33)** | |
| Very easy | 2 (6%) |
| Easy | 14 (42%) |
| Not easy/not hard | 8 (24%) |
| Hard | 6 (18%) |
| Very hard | 2 (6%) |
| Not answered | 1 (3%) |

10 HIV-1 positive and 21 HCV RNA positive ssDBS samples were compared to their corresponding labDBS samples. In this analysis, we used the results of ssDBS eluted in L6-buffer, and their corresponding labDBS. The median time between sampling and elution of the ssDBS was 4 days (IQR 2.5–6.5) for HCV RNA positive samples compared to 3 days (IQR 1.5–5) for corresponding labDBS. For HIV-1 RNA positive ssDBS samples the median time between sampling and elution was 5 days (IQR 2–8) compared to 1 day (IQR 1–1.5) for corresponding labDBS. The correlation between ssDBS and labDBS HCV RNA values was high (r = 0.833, Fig 1A). Comparing the HCV viral loads, the Bland-Altman analysis showed that the mean difference of HCV RNA between labDBS and ssDBS was 0.3 $\log_{10}$ IU/mL (95%CI = 0.0, 0.6) and the limit of agreement was between -1.0 to 1.7 $\log_{10}$ IU/mL (Fig 2A).

The detection of HCV RNA in ssDBS (eluted in L6-buffer) showed a minimal difference in sensitivity, compared to LabDBS (eluted in L6-buffer). Both methods had high sensitivity: 95.7% (95% CI: 78.1–99.9) for ssDBS and 96.4% (95% CI: 81.7–99.9) for LabDBS.

A strong correlation was also found between ssDBS and labDBS HIV-1 RNA values (r = 0.857, Fig 1B). In the Bland-Altman analysis, the mean difference between labDBS and ssDBS HIV-1 viral load measurements was 0.1 $\log_{10}$ copies/mL (95% CI = -0.2, 0.5) and the limit of agreement was between -0.8 to 1.1 $\log_{10}$ copies/mL (Fig 2B).

The detection of HIV RNA in DBS (eluted in L6-buffer) showed a sensitivity of 100% for both ssDBS (95%: 69.2–100) and LabDBS (95%CI: 73.5–100).

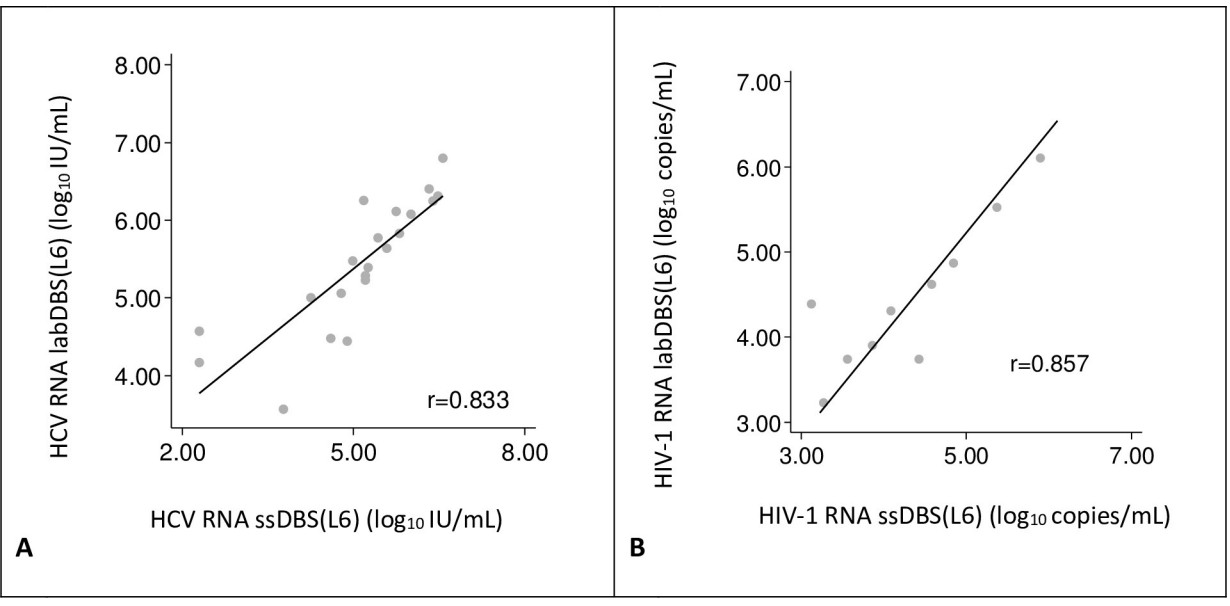

**Fig 1. Correlation between self-sampled dried blood spots (ssDBS) and lab-sampled dried blood spots (labDBS) viral load measurements.**
Individual viral load measurements are plotted (gray dots). The solid line indicates the linear regression line between the two samples. (A) **HCV loads**. The linear relationship between the samples is defined as: $\log_{10}$ IU/mL labDBS HCV = 0.595 x ($\log_{10}$ IU/mL ssDBS) + 2.402. (B) **HIV-1 loads.** The linear relationship between the samples is defined as: $\log_{10}$ copies/mL labDBS HIV-1 = 0.833 x ($\log_{10}$ copies/mL ssDBS) +0.857.

## Comparison of plasma and labDBS viral loads

All 39 plasma samples provided quantifiable results, while HCV RNA could not be detected for 1 labDBS sample with a plasma HCV viral load of 2.98 $\log_{10}$ IU/mL. In total 26 HCV positive and 11 HIV-1 positive labDBS samples (eluted in L6) were compared to their corresponding plasma samples. Viral load values measured in labDBS and plasma are shown in Fig 3. A strong correlation was found between labDBS and plasma HCV RNA values (r = 0.958, Fig 3A) with a linear relationship between the samples. The correlation found between labDBS and plasma HIV-1 values was high (r = 0.844, Fig 3B).

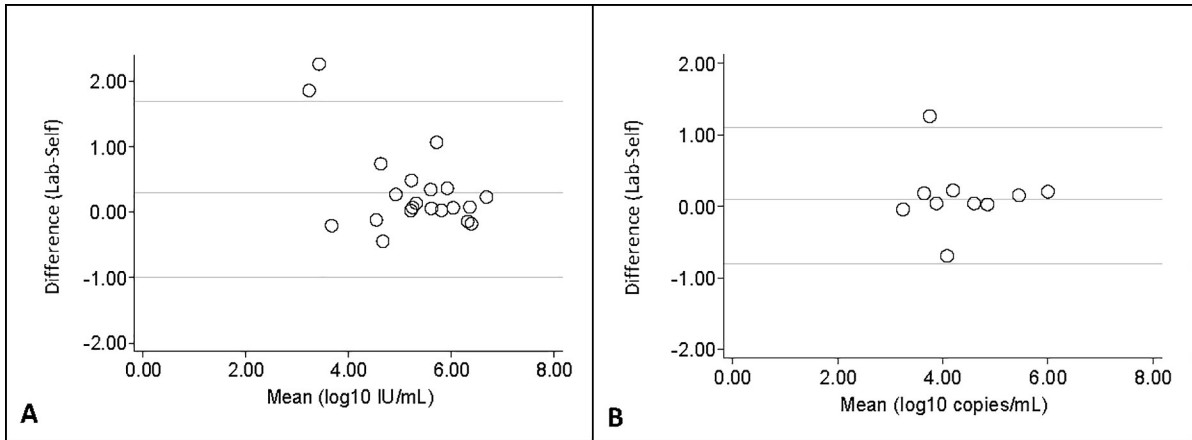

**Fig 2. Bland-Altman analysis comparing the labDBS and ssDBS viral load measurements to their mean.** The solid line in the middle represents the mean difference between labDBS (laboratory sampled dried blood spots) and ssDBS (self-sampled dried blood spots) viral loads, while the lower and upper lines are for the limits of agreement (± 2 standard deviations). (A)**HCV loads**: 2 values lie outside the limits of agreement; (B) **HIV-1 loads**: one value lies outside the limits of agreement.

## Evaluation of elution buffers

A total of 36 labDBS samples (25 of HCV-RNA positive patients and 11 of HIV-1 RNA positive patients) were eluted in L6-buffer as well as in PBS. In 24 out of 25 labDBS samples, HCV RNA was detected and in 11 labDBS samples, HIV-1 RNA was detected; resulting in 35 quantifiable viral load measurements for comparison.

Viral load measurements in L6-buffer eluted labDBS were compared to their corresponding labDBS samples eluted in PBS (Fig 4). A strong correlation was found between HCV RNA loads measured in L6 and PBS eluates (r = 0.913, Fig 4A). In the Bland-Altman analysis, the mean difference between HCV viral loads measured in PBS and L6 eluates, was -0.7 $\log_{10}$ IU/mL (95% CI = -0.8, -0.5), showing higher viral load results in the L6 eluted samples.

The correlation found between HIV-1 RNA viral loads measured in L6 and PBS-eluates was also high (r = 0.835, Fig 4B). From the Bland-Altman analysis, the mean difference between HIV-1 viral loads in PBS and L6 eluates was -0.6 $\log_{10}$ copies/mL (95% CI = -1.0,-0.3). Again, showing higher viral load results in the samples eluted in the L6 buffer.

## RNA stability in DBS

Two to three consecutive viral loads were measured over time of 23 HCV and 8 HIV-1 RNA positive labDBS samples. The samples were stored for a maximum of 21 days at RT after sampling. Samples were measured at t0 (stored <7 days) and/or at t1 (stored 7–13 days) and/or t2 (stored > 14 days). Fig 5 shows the viral load difference between plasma and labDBS plotted over time for each individual sample and the average difference. No statistically significant change in viral load difference was found over the time period up to 21 days, for either HIV-1 or HCV viral loads.

## Discussion

We have taken a comprehensive approach to evaluating dried blood spot self-sampling at home for the detection of HCV RNA and HIV-1 RNA. We compared the detection of HCV

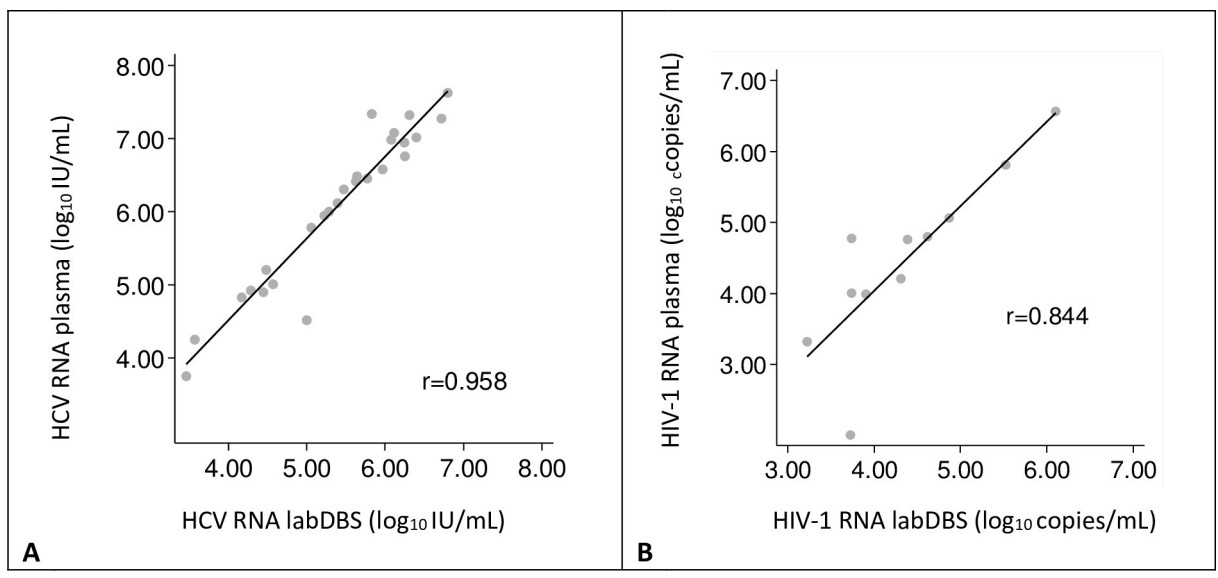

**Fig 3. Correlation between laboratory sampled dried blood spots (labDBS) and plasma viral load measurements.** Individual viral load measurements are plotted (gray dots). The solid line indicates the linear regression line between the two samples. (A) **HCV loads**. The linear relationship between the samples is defined as: $\log_{10}$ IU/mL plasma HCV = 1.119 x ($\log_{10}$ IU/mL labDBS) + 0.036. (B) **HIV-1 loads.** The linear relationship between the samples is defined as: $\log_{10}$ copies/mL plasma HIV-1 = 1.193 x ($\log_{10}$ copies/mL labDBS) - 0.738.

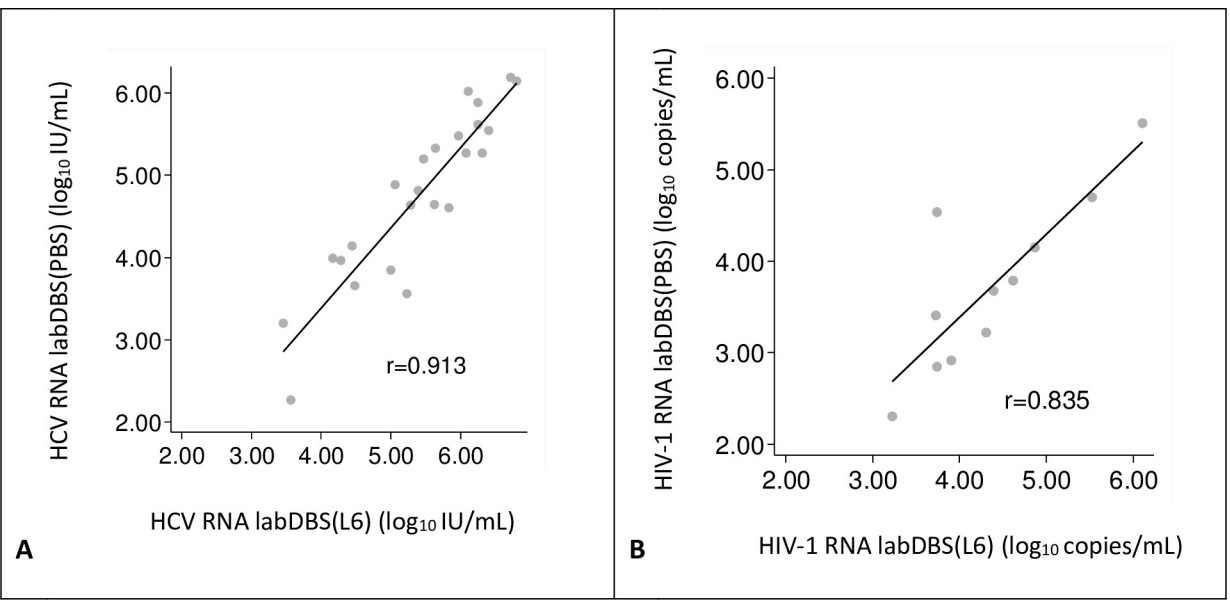

**Fig 4. Correlation between viral load measurements of laboratory sampled DBS (labDBS) eluted in L6-buffer and PBS.** Individual viral load measurements are plotted (gray dots). The solid line indicates the linear regression line between the two samples. (A) **HCV loads**. The linear relationship between the samples is defined as: $\log_{10}$ IU/mL labDBS (PBS) = $0.978*\log_{10}$ IU/mL labDBS (L6) –0.534. (B) **HIV-1 loads**. The linear relationship between the samples is defined as: $\log_{10}$ copies/mL labDBS (PBS) = $0.910*\log_{10}$ copies/mL labDBS (L6)– 0.252. DBS (PBS) = DBS eluted in PBS, DBS (L6) = DBS eluted in L6.

RNA and HIV-1-RNA in plasma, labDBs, and ssDBS, use of two elution buffers and assessed the quality of ssDBS and user-friendliness of self-sampling. We clearly show that HCV RNA and HIV-1 RNA can reliably be detected in self-sampled DBS. Furthermore, the use of L6 rather than PBS as elution buffer resulted in higher viral load values, in particular for HCV, which suggests that more RNA is eluted, resulting in increased sensitivity compared to PBS as

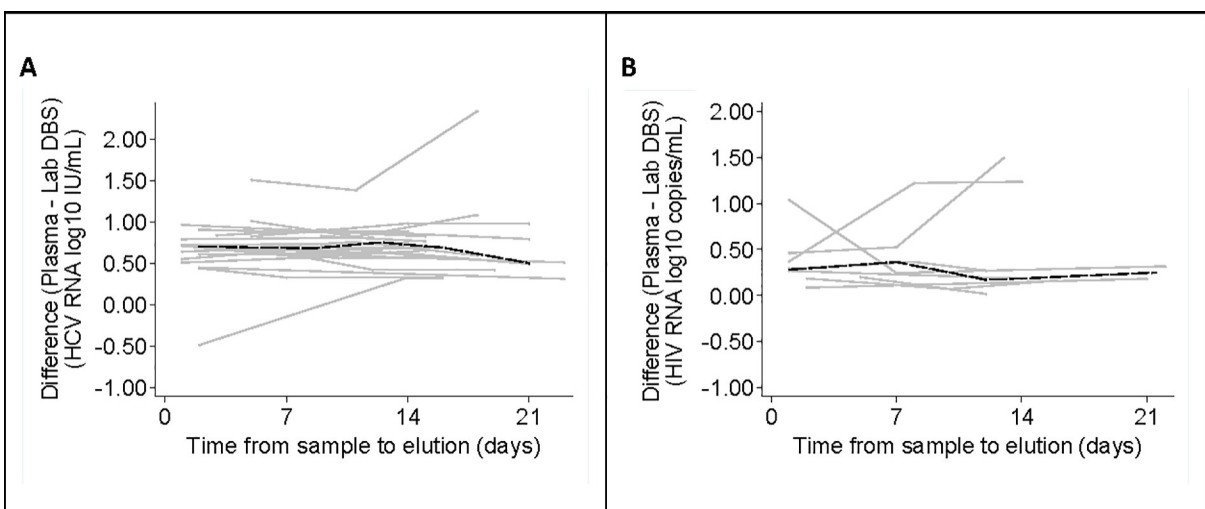

**Fig 5. Difference in viral load measurements between plasma and labDBS samples stored for up to 21 days.** Viral load measurements of labDBS are plotted over time (gray lines). The black dashed line represents the average difference over time. (A) **HCV loads**: The average change in difference over time is $0.002 \log_{10}$ IU/mL (95% CI = -0.005, 0.009), p = 0.6. (B) **HIV-1 loads**: The average change in difference over time is $0.004 \log_{10}$ copies/mL (95% CI = -0.018, 0.026), p = 0.7.

elution buffer. Finally, we showed that HCV and HIV-1 RNA in DBS stored at room temperature are stable for a period of up to 21 days.

Although the majority (97%) of the patients used the paper sampling instructions, and found them helpful, only 36% returned the requested 5 spots with good quality. The main problem we observed with samples of low quality was that the sampled spots were too small. This was unexpected as we had taken the same approach as Hoeller et al., who reported that the large majority (98%) of ssDBS samples were of very good quality [20]. Similar to their approach, we provided patients with both paper and video instructions and assessed samples according to the same quality criteria. The low number of good quality spots in our study may be explained by the fact that about half of the patients did not use the video instructions and some patients experienced performing a finger prick and filling the circles correctly as (very) hard. Even though a substantial number of the ssDBS samples were not rated as 'good quality', HCV RNA could be detected in all but one ssDBS sample.

To our best knowledge, there are no other reports have been published comparing different buffers to optimise DBS elution protocols for the detection of HCV and/or HIV-1 RNA. Our comparison of two different elution buffers showed that L6 buffer gave more sensitive results with a 0.7 $\log_{10}$ IU/mL mean difference in measured HCV-RNA viral loads compared to PBS. This finding has resulted in adjusting the elution protocol of our testing service to the use of L6-buffer and thus lowering the detection limit of the HCV-RNA test. The L6-buffer also gave more sensitive results for HIV-RNA measurements, but with a 0.6 $\log_{10}$ copies/mL higher level with L6 versus PBS eluted samples.

The utility of DBS as a sampling technique for diagnosis of HCV has been evaluated before and shown to be a good tool for hepatitis C screening and diagnosis [14, 23]. It has been demonstrated that measuring HCV RNA viral loads in DBS is specific and sensitive and strongly correlates with serum HCV-RNA measurements [9]. From the data obtained in our study, we also observed that HCV-RNA viral loads quantified using labDBS and plasma samples showed strong correlation. Viral loads measured in labDBS were lower than those measured in plasma, while one labDBS sample with a low HCV-RNA load gave a negative result. This false negative result can be explained by the reduced sensitivity when using DBS because of the lower input of material. The plasma result of the negative labDBS sample was 3.0 $\log_{10}$ IU/mL, which falls within the range of the previously reported detection limits of 2.2–3.4 $\log_{10}$ IU/mL [8–11]. Similarly, we observed a strong correlation between labDBS and plasma samples for HIV-1 RNA measurements and a decreased sensitivity of DBS compared to plasma, in line with previous studies [12, 24].

As we observed a high agreement between ssDBS and labDBS HCV and HIV-1 RNA viral loads, this study shows, for the first time, that HCV RNA and HIV-1 RNA can adequately be detected in a self-sampled DBS at home. SsDBS can also be used to measure HCV RNA with high sensitivity (96.4%). This result falls in line with a systematic review of the diagnostic accuracy of detecting HCV RNA using DBS, reporting a pooled sensitivity of 98% (95% CI: 95–99) [25].

Furthermore, our data indicate that HCV and HIV-1 RNA stay stable in DBS at RT for up to 3 weeks as RNA levels in DBS measured over time did not change, confirming a previous report [15]. Since time from DBS self-sampling to analysis is approximately 1 to 1.5 weeks, including drying time and transport of the sample by regular mail, this result implies that self-sampled DBS use in a real-life setting is feasible.

Our study had some limitations. As the ssDBS were sampled at home, by patients themselves, we did not know the exact volume of blood applied to the DBS. We did assess this qualitatively, but we did not measure the surface area of the spot. The small difference between labDBS and ssDBS viral load measurements could be caused by a difference in sample volume.

Furthermore, we could not assess if patients dried the DBS according to instructions or the time between sampling and posting of the sample. However, by testing the ssDBS and comparing the results to labDBS, we have shown that comparable results can be obtained and thus sampling and posting instructions were assumed to be followed correctly. Although the intended use of the home-based testing service in our project was to diagnose acute infections in MSM at risk of HCV, we did not exclusively include patients with acute HCV infections. One third of the HCV viremic participants had an acute infection. Compared to the chronic phase, where HCV RNA levels tend to remain stable or increase over time [26], early infections are characterised by viral load fluctuations and low HCV RNA levels ($<5 \log_{10}$) [27–29]. Early infections could, therefore, be missed at an early stage of the infection when using the DBS technique. In this study, one patient had a low HCV viral load of $3.0 \log_{10}$ IU/mL, which could not be detected in DBS. As DBS is less sensitive than a plasma sample test, it is advisable to repeat a DBS test after 2–4 weeks if acute HCV infection is suspected.

The main benefits of ssDBS HCV-RNA testing are that DBS sampling is easy and can be performed anonymously at home without the need of any additional resources. This could increase access to testing for the group that does not regularly attend healthcare services, is rarely screened for HCV or does not want to disclose their (sexual) risk behaviour to a health professional. We expect this group to be small in the Netherlands. However, the number of individuals who remain undetected and are unaware of their HCV-status is hard to estimate. We intend to reach these individuals with the NoMoreC project. A home sampling testing service may lower barriers to testing, increase testing frequency and thus limit onward transmission. Takano et al. showed that ssDBS for HIV testing is an acceptable method for MSM in Tokyo and can improve access to testing for MSM who live in rural areas [21]. Coats and Dillon found evidence that DBS may increase the frequency of HCV testing [30]. In addition, the use of DBS-based testing can improve the efficiency of testing in high endemic countries and for hard-to-reach populations. Problems with completing the conventional two-step diagnostic process (serology and HCV-RNA confirmatory testing on a follow-up visit) and the asymptomatic nature of early HCV infection are two important factors that hinder early diagnosis [31]. Therefore, simplified diagnostic strategies are urgently needed and should be increased in health facilities and community-based settings. DBS-sampling is well-suited for mobile and outreach testing programmes and has been accepted by WHO as an alternative approach to facilitate access to testing, increase testing rates and reduce loss to follow up [32]. DBS sampling for HCV and HIV diagnosis is also valuable in settings where there is a lack of access to nearby laboratory facilities for viral load measurements, or where timely delivery of specimens to a laboratory cannot be guaranteed.

In conclusion, self-sampling of DBS at home is a suitable technique for diagnosing HCV and HIV-1 infections. The implementation of this one-step diagnostic approach is feasible and can be used to help scale-up HCV and HIV screening of hard-to-reach, at risk populations as, well as in high endemic countries.

## Supporting information

**S1 Data.**
(DTA)

## Acknowledgments

We thank all patients that participated in the study. We thank the following clinical staff of the infectious diseases and hepatology departments, for their assistance in patient recruitment:

Jeltje Helder, Frank Pijnappel, Olivier Richel, Gonneke Hermanides and Esmerij van der Zanden.

## Author Contributions

**Conceptualization:** Tamara Prinsenberg, Janke Schinkel.

**Data curation:** Tamara Prinsenberg, Sjoerd Rebers.

**Formal analysis:** Tamara Prinsenberg, Anders Boyd, Janke Schinkel.

**Funding acquisition:** Maria Prins, Marc van der Valk, Janke Schinkel.

**Methodology:** Tamara Prinsenberg, Janke Schinkel.

**Writing – original draft:** Tamara Prinsenberg.

**Writing – review & editing:** Tamara Prinsenberg, Sjoerd Rebers, Anders Boyd, Freke Zuure, Maria Prins, Marc van der Valk, Janke Schinkel.

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
