## [Decision Letter · Decision Letter 0]

19 Dec 2019

PONE-D-19-23054

Dried blood spot self-sampling at home is a feasible technique for hepatitis C RNA detection

PLOS ONE

Dear Ms. Prinsenberg,

Thank you for submitting your manuscript to PLOS ONE. After careful consideration, we feel that it has merit but does not fully meet PLOS ONE’s publication criteria as it currently stands. Therefore, we invite you to submit a revised version of the manuscript that addresses the points raised during the review process.

Your manuscript was reviewed by 2 experts in the field. Both expressed significant concerns and identified several important problems in your paper that require your attention. Please review the attached comments and provide point-by-point responses.

We would appreciate receiving your revised manuscript by Feb 02 2020 11:59PM. To enhance the reproducibility of your results, we recommend that if applicable you deposit your laboratory protocols in protocols.io, where a protocol can be assigned its own identifier (DOI) such that it can be cited independently in the future. For instructions see: http://journals.plos.org/plosone/s/submission-guidelines#loc-laboratory-protocols

We look forward to receiving your revised manuscript.

Kind regards,

Yury E Khudyakov, PhD

Academic Editor

PLOS ONE

Journal Requirements:

1.

3. Thank you for stating the following in the Competing Interests section: "I have read the journal's policy and the authors of this manuscript have the following competing interests:TP, FZ and MP report  speaker fees  and grants from: Gilead Sciences, MSD and AbbVie paid to their institute. AB reports grants from ANRS and SIDACTION, outside the submitted work. SR has no relevant conflicts of interest to report.  MvdV’s institute received grants and speaker fees from: Abbvie, Gilead, Johnson & Johnson, MSD, ViiV, outside the submitted work. JS reports non-financial support from ROCHE Diagnostics, during the conduct of the study and grants from: Gilead Sciences, MSD, Abbvie, outside the submitted work."

We note that one or more of the authors are employed by a commercial company.

Please also provide an updated Competing Interests Statement declaring this commercial affiliation along with any other relevant declarations relating to employment, consultancy, patents, products in development, or marketed products, etc.  

Reviewers' comments:

Reviewer's Responses to Questions

**Comments to the Author**

1. Is the manuscript technically sound, and do the data support the conclusions?

Reviewer #1: Yes

Reviewer #2: Partly

2. Has the statistical analysis been performed appropriately and rigorously? 

Reviewer #1: Yes

Reviewer #2: No

3. Have the authors made all data underlying the findings in their manuscript fully available?

Reviewer #1: Yes

Reviewer #2: No

4. Is the manuscript presented in an intelligible fashion and written in standard English?

Reviewer #1: Yes

Reviewer #2: Yes

5. Review Comments to the Author

Reviewer #1: This manuscript is well written and has clear objectives and the methodology and analysis are clear and consise to answer their questions. However, there are a few points that the authors can clarify more as to the use of this type of testing. Especially since they note that most new cases are in MSM populations, mostly those co-infected with HIV or possibly those on PrEp who will be regularly seen based on their medication needs. They do not address how co-infected individuals may react on these tests or mention it. This seems like an important consideration. I have made note of a few additional comments and suggestions for this manuscript. But in general, the idea of home based sampling is a positive move especially for marginalizing populations who may be more weary of going to a health facility.

Abstract:

Line 36 - PBS - is this supped to be DBS if not, please define PBS in abstract.

Methods:

Line 110 - Were the participants supposed to put the desiccant in the sachet as well? As described in the LabDBS procedures

Line 135 - Authors mention cutting two full circles into small pieces - was the full 2 circles used in the elution or were these separated into different eppendorfs for elution? If separated, would be helpful for more specifics on size. Also noting why they were cute into small pieces would be useful.

Line 162: The same process as described above - with two spots? Clarify.

Line 176: en - is this an error? should it be and?

Results:

Table 2: Would suggest the subsets of used (helpful/not helpful, clear, ect) be the percent of those saying they used it. Currently, it looks like the percents are of the total group, but this includes also those who did use those methods. For example in the video instructions, 17 people used them - 15 found them helpful, 2 did not - but then the % presented is a part of the total 33 not the 17 that used them.

Discussion:

This paper starts the introduction discussing the role of HCV in HIV1 infected MSM - indicating there may be a high level of co-infection in this population, however, the authors look at each disease individually, either HIV only or HCV only - the reason for this was not described - this could be listed as a limitation or in the methodology. Was this done due to cross-reactivity? As it is thought that most HCV cases are now in MSM (both HIV infected and those on PrEP - who regularly are seeking medical care and being tested) as opposed to IDUs at this point - is there an idea of how many of the new HCV cases will be in harder to reach populations.

Reviewer #2: The aim of this article was to evaluate whether dried blood spot self-sampling (ssDBS) at home had a good reliability for detection of hepatitis C virus (HCV) RNA. The question is fully relevant since alternative strategies are needed to provide a broad access to HCV testing in several different circumstances. The results are interesting and convincing. However I feel that there are several weaknesses that I will describe below.

- The question focused on HCV RNA detection. Therefore, introducing data on HIV RNA is confusing and does not bring additional information. In addition, the distribution of virus loads in chronically infected individuals is highly different between HCV and HIV. Everything related to HIV should be removed.

- Similarly, a limited amount of data on HCV antigen (HCV Ag) is presented. As the authors stated themselves, additional studies are needed to explore this further (lines 402-406). The parts related to HCV Ag should be deleted.

- Correlations are not enough. The authors should provide data on the sensitivity (at least on labDBS).

Specific comments.

- Abstract: L6 should be introduced. We know what it means only after reading the full text.

- References 1 & 2: more recent papers should be included.

- DBS processing: two spots were used. What was the diameter/size of the spots? How were the spots cut? What measures to avoid cross-contaminations? More details should be provided.

- Line 138: 500 g GuSCN? Please confirm.

- Line 143: 14,000 rpm? (see: 1,400 rpm on line 149). Please confirm.

- Line 146: BSA buffer was already described before (see line 137).

- Table 1 (and in other parts of the paper): it is not clear if the patients were mono-infected (HCV or HIV) or co-infected => See line 242: 34 ssDBS (in table1: 27 HCV and 12 HIV).

- Comparison of plasma and labDBS viral load: the correlation was high, that’s fine. But the authors should also estimate the sensitivity.

- Lines 358-359: “Furthermore … lower detection limit…”. Please precise the value of the detection limit. Same lines 288-389 (“decreased sensitivity on DBS”) => it must be determined precisely.

- Line 370, “… regardless of the quality of the spot”. It is not possible to claim that. The smaller is the spot, the lower will be the sensitivity!!

- Line 396: The authors should include ref (Takano et al., BMC Infect Dis 2018, 18:627), who demonstrated the feasibility of using ssDBS and postal service for HIV prevalence studies.

6. PLOS authors have the option to publish the peer review history of their article (what does this mean?). If published, this will include your full peer review and any attached files.

Reviewer #1: Yes: Nicole Hoff

Reviewer #2: No

---

## [Author Response · Author response to Decision Letter 0]

5 Mar 2020

Response to reviewer’s comments

Title: Dried blood spot self-sampling at home is a feasible technique for hepatitis C RNA detection

Reviewers comments in black our response is given in blue font.

Reviewer #1: 

This manuscript is well written and has clear objectives and the methodology and analysis are clear and concise to answer their questions. However, there are a few points that the authors can clarify more as to the use of this type of testing. Especially since they note that most new cases are in MSM populations, mostly those co-infected with HIV or possibly those on PrEp who will be regularly seen based on their medication needs. They do not address how co-infected individuals may react on these tests or mention it. This seems like an important consideration. I have made note of a few additional comments and suggestions for this manuscript. But in general, the idea of home based sampling is a positive move especially for marginalizing populations who may be more weary of going to a health facility.

We thank the reviewer for this positive response. 

In our study about one third (11/39) of our participants were HIV-HCV co-infected individuals. The reaction of the co-infected persons on the home-sampled test will be part of another study where we will evaluate the NoMoreC project. The NoMoreC project offers the home-sampled HCV RNA test in a real life setting to MSM at risk of HCV. The experiences and reactions of users of the NoMoreC HCV-RNA testing service are currently being collected through questionnaires and interviews. We will report these in the future. We focused on the feasibility of self-sampling DBS but did however include the experience of the sampling method in this manuscript.

Abstract:

1. Line 36 - PBS - is this supposed to be DBS if not, please define PBS in abstract.

It is supposed to be PBS (phosphate-buffered saline). This is now defined in the abstract and introduced in line 27 of the revised clean manuscript as one of the elution buffers.

Methods:

2. Line 110 - Were the participants supposed to put the desiccant in the sachet as well? As described in the LabDBS procedures

Yes, now modified in manuscript (line 111).

3. Line 135 - Authors mention cutting two full circles into small pieces - was the full 2 circles used in the elution or were these separated into different eppendorfs for elution? If separated, would be helpful for more specifics on size. Also noting why they were cute into small pieces would be useful.

More details are given on how and why the spots were cut up into small pieces (line 137-139).

We did not measure the size of the self-sampled spots, which is a limitation of our study. We mention this in the discussion. 

The volume of the laboratory spots were, however, known: 60 µL of EDTA whole blood per circle. This is described in the methods section under Plasma and labDBS collection (line 130). 

4. Line 162: The same process as described above - with two spots? Clarify. 

HCV RNA measurement for DBS has now been described more clearly: 650 µL DBS eluate from 2 spots was used (line 151 and 154). 

5. Line 176: en- is this an error? Should it be and? 

Yes, this was a typo, now corrected in the manuscript (168).

Results:

6. Table 2: Would suggest the subsets of used (helpful/not helpful, clear, ect) be the percent of those saying they used it. Currently, it looks like the percents are of the total group, but this includes also those who did use those methods. For example in the video instructions, 17 people used them - 15 found them helpful, 2 did not - but then the % presented is a part of the total 33 not the 17 that used them. 

Thank you for this suggestion. Indeed, it makes more sense to give the percentages of participants who actually used the instructions instead of the whole group. The table has been modified accordingly (lines 208-226).

Discussion:

7. This paper starts the introduction discussing the role of HCV in HIV1 infected MSM - indicating there may be a high level of co-infection in this population, however, the authors look at each disease individually, either HIV only or HCV only - the reason for this was not described - this could be listed as a limitation or in the methodology. Was this done due to cross-reactivity? As it is thought that most HCV cases are now in MSM (both HIV infected and those on PrEP - who regularly are seeking medical care and being tested) as opposed to IDUs at this point - is there an idea of how many of the new HCV cases will be in harder to reach populations.

Thank you for this comment. We included participants who were expected to be HCV or HIV viremic. This was the only inclusion criterion. Some (9/27) of the HCV-positive patients had controlled HIV-infection and thus an undetectable HIV viral load. Only one patient was co-infected with detectable HIV and HCV viral loads at inclusion. To be more thorough, we have now given the numbers of mono-infected and co-infected patients in the result section (lines 181-184), including the numbers of those with undetectable HIV viral load. We also provided information on measured samples according to mono and co-infection in Table 1 (line 191, page 9).

Cross-reactivity is not an issue with RNA measurements as virus-specific RNA is amplified and subsequently measured. 

With regard to the number of HCV cases in harder-to-reach populations, the large majority of MSM in the Netherlands who are HIV-positive or HIV-negative on PrEP are indeed in care. We have developed a service that allows MSM at risk to test themselves more frequently at their own convenience, thus detecting acute infection earlier and preventing onward transmission sooner. This home self-sampling HCV service is in addition to their regular control visits at the clinic. We have added the advantages of home testing in the introduction (lines 61-64). In the Netherlands, the number of new HCV cases that are hard to reach is expected to be quite small, although this is difficult to estimate (added to the discussion lines 398-400). Data from the Netherlands show that the number of new HCV infections among PWID has remained nearly zero over the last decade as we described in the introduction of the original manuscript (line45 in the revised manuscript). With our NoMoreC project, we intend to reach those individuals who remain undetected or are unaware of their HCV-status (added to the discussion in lines 400-401) It is expected that the ssDBS technique could be used in other countries with hard-to-reach/hard-to-serve populations, which we refer to in our discussion of the original manuscript (lines 406 & 418 revised manuscript).

Reviewer #2: The aim of this article was to evaluate whether dried blood spot self-sampling (ssDBS) at home had a good reliability for detection of hepatitis C virus (HCV) RNA. The question is fully relevant since alternative strategies are needed to provide a broad access to HCV testing in several different circumstances. The results are interesting and convincing. However I feel that there are several weaknesses that I will describe below.

1. The question focused on HCV RNA detection. Therefore, introducing data on HIV RNA is confusing and does not bring additional information. In addition, the distribution of virus loads in chronically infected individuals is highly different between HCV and HIV. Everything related to HIV should be removed. 

The main aim of our study was to evaluate the feasibility of self-sampling DBS at home. With our study, we wanted to demonstrate that people are capable of self-sampling DBS using the instructions provided, resulting in concordant diagnostic test results compared with a labDBS, regardless of pathogen tested. Omitting one third of the study results, which are potentially relevant to others who want to implement self-sampling by DBS, seems undesirable. 

Therefore, we kept all data on HIV in the manuscript, as we believe it provides additional value for others who aim to develop home-based testing methods.

2. Similarly, a limited amount of data on HCV antigen (HCV Ag) is presented. As the authors stated themselves, additional studies are needed to explore this further (lines 402-406). The parts related to HCV Ag should be deleted. 

We explored the use of HCV core antigen mainly because we felt this this could be a cost-saving alternative to HCV-RNA detection. We agree with the reviewer and as the numbers were quite small, we have removed the parts related to HCV antigen from the manuscript

3. Correlations are not enough. The authors should provide data on the sensitivity (at least on labDBS). 

We thank the reviewer for his/her suggestion and have calculated the sensitivity for both LabDBS as ssDBS, which were added to the results (line 228, 229, 251,252) and discussion (line 350)

Specific comments.

4. Abstract: L6 should be introduced. We know what it means only after reading the full text. 

We have now introduced both buffers (PBS and L6) in line 28.

5. References 1 & 2: more recent papers should be included.

In the Netherlands, the incidence among people who inject drugs has been practically reduced to zero. This is still the case. We have now included the most recent National report on sexually transmitted infections in the Netherlands from the National Institute for Public Health and the Environment (RIVM): reference 3 (line 45). This report gives the number of acute hepatitis C infections per year by injection drug use from 2009 up to 2018, during which no or almost no cases of HCV were observed.

6. DBS processing: two spots were used. What was the diameter/size of the spots? How were the spots cut? What measures to avoid cross-contaminations? More details should be provided.

See our response to comment 3 of reviewer 1. More details are now given in the manuscript.

7. Line 138: 500 g GuSCN? Please confirm. 

The L6-buffer does indeed contain 500 g Guanidinium thiocyanate (GuSCN). 

In the article describing how to prepare the L6- buffer (ref. 21), 120 grams of GuSCN is dissolved in 100 ml of 0.1M Tris HCl. We prepared a larger volume and dissolved 500 grams of GuSCN in 416.7 mL of 0.1M Tris-HCL.

8. Line 143: 14,000 rpm? (see: 1,400 rpm on line 149). Please confirm.

14,000 rpm is correct (line 146)

9. Line 146: BSA buffer was already described before (see line 137).

Modified accordingly

10. Table 1 (and in other parts of the paper): it is not clear if the patients were mono-infected (HCV or HIV) or co-infected => See line 242: 34 ssDBS (in table1: 27 HCV and 12 HIV).

Table 1 has been modified: the number of co-infected and mono-infected patients is now provided (line 191, page 9). 

11. Comparison of plasma and labDBS viral load: the correlation was high, that’s fine. But the authors should also estimate the sensitivity.

The sensitivities for viral load measurements in labDBS and ssDBS have been estimated and are included in the results for HCV RNA (line 248-249) and HIV RNA (lines 271-272)

12. Lines 358-359: “Furthermore … lower detection limit…”. Please precise the value of the detection limit. Same lines 288-389 (“decreased sensitivity on DBS”) => it must be determined precisely.

We did not measure the limit of detection. Rather, we found that samples eluted in L6 gave a higher viral load measurement compared to PBS. As we could not report the detection limit, we avoided this term and reported the difference in viral loads.

Our goal was not to measure the detection limit, as this has been determined by other groups (references 8-11). 

13. Line 370, “… regardless of the quality of the spot”. It is not possible to claim that. The smaller is the spot, the lower will be the sensitivity!!

We appreciate this comment. There seems to be some confusion regarding the term ‘sensitivity’. Lower spot sizes will indeed decrease ‘analytical sensitivity’, which is defined by the limit of detection. As long as there is enough RNA in the sample, spot sizes should not influence the ‘diagnostic sensitivity’, which is the fraction of true positive RNA results. In this clinical evaluation, the size of the spot did not result in any ‘false-negative’ results. 

To clarify, we have rephrased this sentence (line 327).

14. Line 396: The authors should include ref (Takano et al., BMC Infect Dis 2018, 18:627), who demonstrated the feasibility of using ssDBS and postal service for HIV prevalence studies. 

Thank you for pointing out this interesting article and showing the relevance of testing for HIV using ssDBS. The group used a similar approach, asking participants to provide their own DBS sample and to send it to the laboratory by regular mail for HIV antigen/antibody detection. We have now referred to this study in the introduction (lines 78-79, ref 21). They also showed that ssDBS was a n acceptable method for MSM in Tokyo and can improve the access to testing for MSM who live in rural areas. We have included this in the discussion (lines 402-404)

---

## [Decision Letter · Decision Letter 1]

24 Mar 2020

Dried blood spot self-sampling at home is a feasible technique for hepatitis C RNA detection

PONE-D-19-23054R1

Dear Dr. Prinsenberg,

We are pleased to inform you that your manuscript has been judged scientifically suitable for publication and will be formally accepted for publication once it complies with all outstanding technical requirements.

With kind regards,

Yury E Khudyakov, PhD

Academic Editor

PLOS ONE

Additional Editor Comments (optional):

Reviewers' comments:

Reviewer's Responses to Questions

**Comments to the Author**

1. If the authors have adequately addressed your comments raised in a previous round of review and you feel that this manuscript is now acceptable for publication, you may indicate that here to bypass the “Comments to the Author” section, enter your conflict of interest statement in the “Confidential to Editor” section, and submit your "Accept" recommendation.

Reviewer #1: All comments have been addressed

Reviewer #2: All comments have been addressed

2. Is the manuscript technically sound, and do the data support the conclusions?

Reviewer #1: Yes

Reviewer #2: Yes

3. Has the statistical analysis been performed appropriately and rigorously? 

Reviewer #1: Yes

Reviewer #2: Yes

4. Have the authors made all data underlying the findings in their manuscript fully available?

Reviewer #1: Yes

Reviewer #2: Yes

5. Is the manuscript presented in an intelligible fashion and written in standard English?

Reviewer #1: Yes

Reviewer #2: Yes

6. Review Comments to the Author

Reviewer #1: This is a well written manuscript. The authors have addressed all comments and suggestions sufficiently. I do not have any additional comments.

Reviewer #2: The authors have modified their manuscript according to our comments. I believe that now this work deserves to be published. It could be useful for many public health advisers.

7. PLOS authors have the option to publish the peer review history of their article (what does this mean?). If published, this will include your full peer review and any attached files.

Reviewer #1: No

Reviewer #2: No

---

## [Editor Report · Acceptance letter]

1 Apr 2020

PONE-D-19-23054R1 

Dried blood spot self-sampling at home is a feasible technique for hepatitis C RNA detection 

Dear Dr. Prinsenberg:

I am pleased to inform you that your manuscript has been deemed suitable for publication in PLOS ONE. Congratulations! Your manuscript is now with our production department. 

With kind regards,

on behalf of

Dr. Yury E Khudyakov 

Academic Editor

PLOS ONE